# Protocol for a randomised, multicentre, four-arm, double-blinded, placebo-controlled trial to assess the benefits and safety of iron supplementation with malaria chemoprevention to children in Malawi: IRMA trial

Martin N. Mwangi,[1,2] Glory Mzembe [1,3] Chikondi C. Ngwira,[1] Maclean Vokhiwa,[1,3] Mayamiko D. Kapulula,[1] Leila M. Larson,[4] Sabine Braat [5,6,7] Rebecca Harding [7] Alistair R. D. McLean [6,7] Jena D. Hamadani,[8] Beverley-Ann Biggs,[5,9] Ricardo Ataíde,[5,7] Kamija S. Phiri,[1,3] Sant-Rayn Pasricha[7,9,10]

MNM and GM are joint first authors.

For numbered affiliations see end of article.

**Correspondence to**
Dr Martin N. Mwangi;
mmwangi@true.mw

## ABSTRACT

**Introduction** Approximately 40% of children aged 6–59 months worldwide are anaemic. Iron-containing multiple micronutrient powders (MNPs) and iron supplements (syrup/drops) are used to combat anaemia in children in different parts of the world. However, evidence for functional benefits of iron supplementation in children is scarce, and potential risks remain poorly defined, particularly concerning diarrhoea and malaria. This trial aims to determine if: (1) the efficacy of iron supplements or MNPs (containing iron) given with malaria chemoprevention is superior to malaria chemoprevention alone, or (2) if the efficacy of malaria chemoprevention alone is superior to placebo on child cognitive development.

**Methods and analysis** IRMA is a four-arm, parallel-group, double-blinded, placebo-controlled, triple-dummy, randomised trial in Southern Malawi. The study recruits 2168 infants aged 6 months, with an intervention period of 6 months and a post-intervention period of a further 6 months. Children are randomised into four arms: (1) No intervention (placebo); (2) malaria chemoprevention only; (3) MNPs and malaria chemoprevention; and (4) iron syrup and malaria chemoprevention. The primary outcome, cognitive development (Cognitive Composite Score (CogCS)), is measured at the end of the 6 months intervention. Secondary outcomes include CogCS at a further 6 months post-intervention, motor, language and behavioural development, physical growth and prevalence of anaemia and iron deficiency. Safety outcomes include incidence of malaria and other infections, and prevalence of malaria parasitaemia during and post-intervention period.

**Ethics and dissemination** The trial is approved by the National Health Sciences Research Committee (#19/01/2213) (Malawi) and the Human Research Ethics Committee (WEHI: 19/012) (Australia). Written informed consent in the local language is obtained from each participant before conducting any study-related procedure.

## STRENGTHS AND LIMITATIONS OF THIS STUDY

⇒ The trial setting represents malaria-endemic sub-Saharan African countries where anaemia and malaria coexist, which is an ideal place to study the benefits versus risks of iron supplementation (under malaria chemoprevention) to establish a generalisable clinical rationale.

⇒ The participant screening and inclusion criteria involve all infants within the recruitment age range across a large geographical catchment area which enables the generalisability of the study results to most national programmes in malaria-endemic countries.

⇒ The trial compares two major delivery modes for iron in young children: multiple micronutrient powders and iron syrup which will help identify the ideal and most effective iron source for children.

⇒ The trial uses a functional clinical health outcome—child cognitive development—as its primary outcome, ensuring the trial provides evidence of the functional relevance of universal iron interventions and malaria prevention.

⇒ The trial is powered to detect a small but clinically relevant effect for the primary outcome of child cognitive development; the use of a post-intervention follow-up period enables the assessment of both the immediate and medium-term impact of the intervention.

Results will be shared with the local community and internationally with academic and policy stakeholders.
**Trial registration number** ACTRN12620000386932.

## INTRODUCTION

Anaemia remains a critical global health problem affecting up to 39.8% of children



aged 6–59 months worldwide and 60.2% in Africa.[1–3] In Malawi, 63% of children aged 6–59 months are anaemic, with a peak prevalence of 91% observed among children aged 9–11 months.[4] Iron deficiency accounts for most (60%) of the global burden of anaemia[5] but less so in malaria-endemic areas, where only 32% of childhood anaemia is iron responsive.[1]

WHO anaemia prevention guidelines recommend two universal iron supplementation strategies for children, namely; (1) iron supplement drops or syrups (10–12.5 mg of elemental iron)—given daily to children 6–24 months in areas where the prevalence of anaemia is 40% or higher; and (2) iron-containing multiple micronutrient powders (MNPs): 10–12.5 mg of iron—given daily over 6 months in areas where the prevalence of anaemia is 20% or higher.[6–8] However, no recommendation is given regarding the preferred approach. MNPs are the most common form of iron supplementation, with at least 58 countries reported to have implemented MN home fortification by 2021.[9] The rationale for universal iron supplementation is an understanding that iron supplementation would reduce childhood anaemia and improve child development and other functional health outcomes. However, we recently showed that in a rural South Asian population, neither universal distribution of oral iron supplements nor MNPs meaningfully improved immediate or medium-term child cognitive, language, motor and other developmental outcomes, even though anaemia and iron deficiency were reduced.[10] However, these data were generated from a trial in a non-malaria endemic part of South Asia, where the determinants of anaemia and infection risk are likely very different.

### Benefits of universal iron supplementation programmes

Universal iron supplementation programmes have shown benefits in haematological outcomes. A meta-analysis[11] and a Cochrane review[12] showed improvements in anaemia with daily iron supplementation in children. However, the effects on growth and development are less clear. Few studies have investigated the impact on cognitive development. Only 8 of the 33 studies included in the meta-analysis by Pasricha *et al* assessed cognitive development, and no significant beneficial effect was seen (mean difference 1.65 (95% CI –0.63 to 3.94)), but data were sparse.[11] Subgroup analysis did suggest a benefit of iron supplementation on cognitive development in iron-deficient children. However, this data came from only two studies.[13 14] Only one study in the Cochrane review assessed child development, showing that children receiving MNPs were more likely to walk independently at 12 months of age than those receiving no intervention.[15] There is also little evidence for a beneficial effect on growth[11 12 16]; however, data were limited.[11] WHO has acknowledged the need for further research into the functional outcomes of iron supplementation programmes.[7]

### Risks of iron supplementation

While the evidence for functional benefits of iron supplementation is scarce, there has been emerging evidence of potential harm, particularly regarding increased malaria risk in endemic areas. The seminal Pemba randomised controlled trial (RCT) in Tanzania was stopped early due to increased death, hospitalisation and malaria in iron replete children receiving iron compared with placebo and called for revisions of universal iron supplementation recommendations.[17] In another RCT in Tanzania, multinutrient supplementation (including iron) increased the number of malaria cases by 41% in iron-deficient children.[18] However, a 2016 Cochrane review found iron supplementation safe in areas with adequate malaria prevention and management strategies.[19] In contrast, there was an increased risk in areas where this was not the case.[19] The current recommendation is for universal iron supplementation in malaria-endemic regions with a high prevalence of anaemia only 'in conjunction with public health measures to prevent, diagnose and treat malaria'.[7]

### Iron deficiency protects against malaria

There is evidence to suggest that iron deficiency is protective against malaria. A study of preschool children in Malawi found that iron-deficient children were at lower risk of clinical malaria and parasitaemia than iron-replete children.[20] In vitro studies have also demonstrated that *Plasmodium falciparum* parasites are less effective at invading iron deficient red cells and prefer invading reticulocytes (which are increased in the recovery phase of iron deficiency anaemia when iron supplementation is given).[21] Parasite invasion and growth were reduced in laboratory growth and invasion studies of *P. falciparum* using red cells from anaemic children.[22] The protective effect imparted by iron deficiency in both in vitro studies was reversed when iron supplementation was given.[21 22]

### Anaemia in malaria-endemic areas is less iron-responsive

Malaria is a significant cause of severe anaemia, hospitalisation and death in endemic areas.[23–25] Several aetiologies, including malaria, were identified in a study of Malawian children with severe anaemia. However, iron deficiency was not a common contributor.[23] Similarly, in a study in children from The Gambia and Tanzania,[26] 61.2% of children were anaemic, but only 13% had iron deficiency anaemia.[26] In keeping with this, the anaemia in these areas is less iron-responsive.[11 27] Without clear evidence of the neurodevelopmental benefits of iron supplementation, iron interventions are likely to be extremely expensive public health interventions in terms of cost per disability-adjusted life years (DALYs) averted, especially in sub-Saharan Africa.[28]

### Malaria may be a risk factor for impaired cognitive development

Despite few studies, the majority with relatively small sample sizes and varying methods of assessing cognitive function, evidence suggests that severe and

uncomplicated malaria may be independent risk factors for cognitive impairment.[29] In addition, intermittent preventive therapy in preschool and school-aged children, has improved cognitive performance.[30 31]

### Intermittent preventive treatment to decrease malaria and its sequelae

Intermittent preventive treatment in children (IPTc) involves the empiric treatment of malaria at regular intervals and is a potential strategy to decrease malaria rates and sequelae. A Cochrane review that included seven RCTs evaluated the effect of chemoprophylaxis and intermittent treatment for malaria in preschool children in areas with seasonal malaria and showed a reduction in clinical malaria and moderate-to-severe anaemia.[32] Furthermore, in a meta-analysis, iron supplementation reduced the rate of anaemia by 30% in malaria-endemic areas, compared with 60% in non-endemic regions.[11]

### Dihydroartemisinin-piperaquine as chemoprevention

Recommended treatment for *P. falciparum* malaria is with artemisinin-based combination therapy (ACT). Dihydroartemisinin-piperaquine (DP) is an ACT and is among WHO's five recommended first-line treatments.[33] DP with its long half-life, given monthly, is an attractive option for IPTc. In an RCT in children in Uganda using this regimen, DP reduced clinical malaria by 58% (compared with 28% for trimethoprim-sulfamethoxazole and 7% for sulfadoxine-pyrimethamine) and parasitaemia by 76%. IPTc with DP also reduced moderate-to-severe anaemia by 47%.[34] The administration of DP-IPTc during MNPs/iron supplementation may improve the safety of these interventions. However, there are concerns that malaria prevention may delay the development of immunity and increase malaria risk once IPTc is ceased.[32]

### Iron deficiency and malaria coexist in Malawi

Despite important gains made worldwide in malaria control since 2010, worldwide cases increased in 2016. In Malawi, there were a total of 5.2 million presumed and confirmed malaria cases in 2016, 231 970 more cases than the year prior, resulting from a gradual increase in cases since 2014.[35] In addition, the prevalence of malaria parasitaemia in children under 5 years increased from 28% in 2012 to 33% in 2014. These increases come despite increases in insecticide-treated bed net coverage.[36]

Both malaria and iron deficiency anaemia remain important causes of morbidity and mortality in Malawi. In Malawian children under 5 years of age, malaria is the leading cause of death,[37] and the prevalence of anaemia is 63%.[38] Addressing both of these conditions is challenging. Current WHO recommendations for improving anaemia in malaria-endemic regions are difficult to interpret and implement in countries like Malawi, where malaria preventive measures and treatment are present but by no means universal.

### Why this trial is needed

Several fundamental questions concerning the benefits and safety of iron-intervention programmes in malaria-endemic regions remain unanswered. The IRMA trial will assess functional health outcomes from iron supplementation and identify the ideal mode of iron delivery (MNPs vs iron syrup/drops). It will also allow evaluation of the effect of malaria chemoprevention alone on anaemia and child development.

## OBJECTIVES

The primary objective is to determine (1) if the efficacy of iron supplements or MNPs (containing iron) given with malaria chemoprevention is superior to malaria chemoprevention alone, or (2) if the efficacy of malaria chemoprevention alone is superior to placebo, on child cognitive development (measured using locally adapted Bayley Scales of Infant and Toddler Development third edition (Bayley-III) Cognitive Composite Scores[39] at the end of the 6-month intervention period.

The specific objectives are:
1. Determine if the efficacy of (a) iron (as supplements or MNPs) given with malaria chemoprevention is superior to malaria chemoprevention alone or (b) if the efficacy of malaria chemoprevention alone is superior to placebo on.
   i. Child cognitive development, child motor, language, behavioural development, temperament and physical growth at the end of the 6 months intervention and a further 6 months post-intervention.
   ii. Anaemia (and haemoglobin concentration), iron deficiency (and ferritin) and iron deficiency anaemia after 6 months of intervention and 6 months post-intervention.
2. Compare the effects of iron given as supplements to MNPs on the above outcomes after 6 months of intervention and a further 6 months post-intervention.
3. Determine the safety of iron (as supplements or MNPs) given with malaria chemoprevention on the risk of malaria and other morbidities versus malaria chemoprevention alone or placebo during and post-intervention.
4. Determine the efficacy of malaria chemoprevention alone compared with placebo on clinical malaria, parasitaemia and malaria-specific clinical visits and all-cause sick visits during and after intervention.
5. Determine the safety of iron (as supplements or MNPs) given with malaria chemoprevention compared with placebo on infection risk in young children, during and post-intervention.

### Study design

IRMA is a four-arm parallel-group, double-blinded, placebo-controlled, triple-dummy, individually randomised trial comparing: (1) No intervention, (2) malaria chemoprevention only, (3) MNPs+malaria

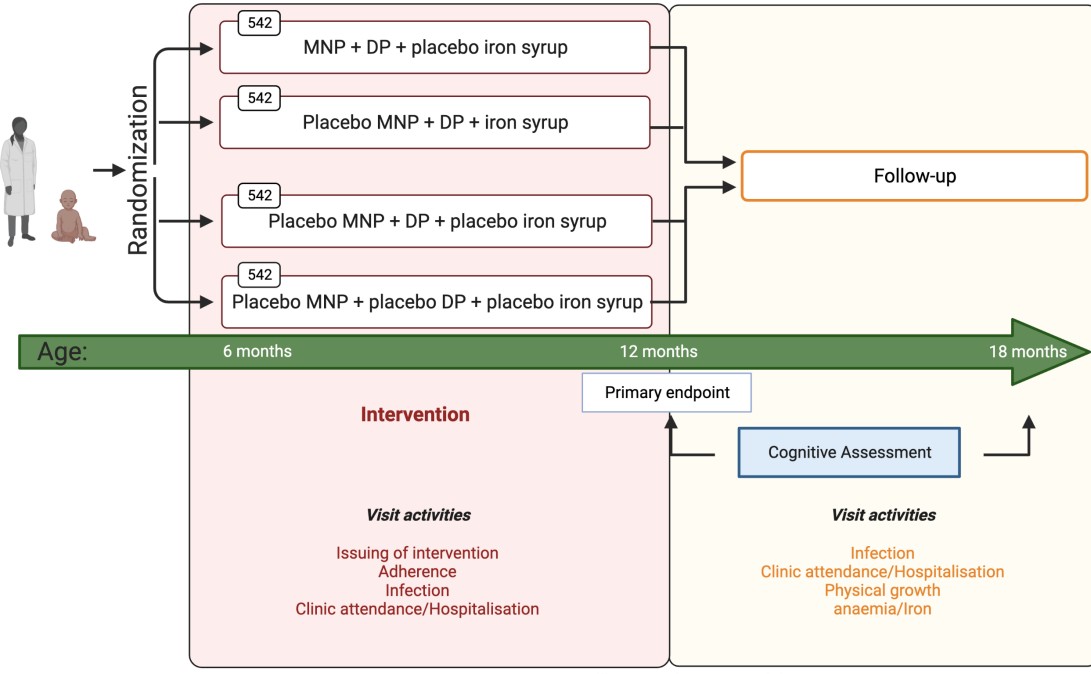

**Figure 1** IRMA trial schema. Overall display of IRMA activities. Intervention—issued monthly, from enrolment at age 6 months to 12 months of age. Primary endpoint—Bayley's assessment conducted at the trial midline; cognitive assessments—Bayley Scales of Infant and Toddler Development third edition. DP, dihydroartemisinin-piperaquine; MNP, iron-containing multiple micronutrient powder.

chemoprevention and (4) iron syrup+malaria chemoprevention at the end of the 6 months intervention and at a further 6 months post-intervention. The trial design is summarised in figure 1.

## Study settings and participants

The trial is set in Chikwawa district, in Southern Malawi, a rural area where >70% of children 6–12 months are anaemic,[40] and where 33% of children under 5 years old have *P. falciparum* parasitaemia.[41] Four health facilities within the district are hosting the trial, namely: Chikwawa District Hospital, Mapelera Health Centre, Mfera Health Centre and Makhwira Health Centre. These are the primary health facilities where children attend routine under-5 clinics and sick visits. Our target participants are all children aged 6 months residing in these health facilities' catchment areas.

## Eligibility criteria

The criteria to select participants considers participants' safety, the generalisability to national programmes and the feasibility of the trial. Children are randomly assigned to a study intervention group only if they meet all the inclusion and none of the exclusion criteria.

## Inclusion criteria

1. Infant aged 6 months (±14 days) at randomisation.
2. Availability of a legally acceptable guardian/representative and providing written consent on the participant's behalf.

## Exclusion criteria

1. Severe anaemia (haemoglobin (Hb)<7 g/dL).
2. Clinical signs of current infective illness (respiratory infection, diarrhoea, malaria); however, children will be screened after recovery and recruited if they meet the age eligibility criteria at the second screening.
3. Severe malnutrition (weight-for-length z-score <–3) or mid-upper arm circumference (MUAC)<11.5 cm.
4. Established haemoglobinopathy diagnosis (eg, sickle cell disease, beta thalassaemia major, haemoglobin E (HbE) thalassaemia).
5. Known congenital anomaly, developmental disorder or severe developmental delay.
6. Children with physical or behavioural problems that will make it impossible to undertake trial assessments.
7. Children of multiple births, for example, twins.
8. Children currently consuming iron supplements or MNPs, for example, from the National MNPs programme.
9. Enrolment in another trial.

## Trial interventions

Participants are randomly assigned to receive one of the following interventions:

1. **No intervention (placebo DP *plus* placebo MNPs *plus* placebo iron syrup)**: Placebo DP (Guilin Pharmaceutical company, China): three consecutive days every 4 weeks for 6 months *plus* placebo MNPs (maltodextrin vehicle only; DSM Nutritional Products, South Africa) daily for 6 months *plus* placebo iron syrup (ACME Laboratories,

Bangladesh) daily for 6 months. Malaria chemoprevention is not given routinely to children under 2 years of age in this setting. Thus, the placebo arm is the current standard of care in the country and no treatment is being withheld from the participants.

2. **Malaria chemoprevention only (active DP** *plus* **placebo MNPs** *plus* **placebo iron syrup**): Active DP (20 mg dihydroartemisinin/160 mg piperaquine; Guilin Pharmaceutical company, China) dosed by body weight (2/16 mg/kg) for three consecutive days every 4 weeks for 6 months *plus* placebo MNPs daily for 6 months, *plus* daily placebo iron syrup for 6 months.

3. **MNP+malaria chemoprevention (active DP** *plus* **active MNPs** *plus* **placebo iron syrup**): Active DP as above, *plus* daily active MNPs containing 10.0 mg iron and 14 other micronutrients (online supplemental file 1); DSM Nutritional Products, South Africa, *plus* placebo iron syrup daily for 6 months.

4. **Iron syrup+malaria chemoprevention (active DP** *plus* **placebo MNPs** *plus* **active iron syrup**): Active DP as above, *plus* placebo MNPs, *plus* active iron syrup containing 10.0 mg ferrous sulphate (ACME Laboratories, Bangladesh) daily for 6 months.

### Randomisation, allocation concealment and blinding
Participants are randomly allocated to one of the four study arms with 1:1:1:1 allocation via a computer-generated randomisation schedule of randomly permuted blocks stratified by sex and health facility/study sites (Chikwawa District Hospital, Mapelera health centre, Mfera health centre and Makhwira Health Centre) to achieve a balance between the intervention arms. An independent statistician from the University of Melbourne (Australia) generated the randomisation list.

An independent trial pharmacist pre-packed and pre-labelled participant packs containing all three investigational products based on the randomisation list. Each participating child who meets the inclusion criteria is sequentially allocated to the next available study number indicated on the randomisation list. The participant is then given a pre-labelled pre-packed Ziplock bag containing the study medication corresponding to the study number assigned to them. Hence, participants and their families and all study team members involved in collecting and analysing outcome data, including field workers, developmental assessors, laboratory scientists, statisticians and investigators, are blinded to the treatment allocation until the database is locked and before unblinding.

### Recruitment and visits
#### Patient and public involvement
All levels of the local leadership (head men/women, subchiefs, chiefs, Community Health Action Groups (CHAGS), Health Surveillance Assistants (HSAs) and District Health Assembly) are continuously engaged by the trial's Community Engagement Team (CET). CHAGs and HSAs were engaged to ensure that we reach the whole community via it's resident health 'gate-keepers'. Additionally, the CET conducted community sensitisation meetings to explain the research objectives, what it means to participate, benefits for the individual or the community and overall contribution to science. Finally, community members are regularly invited to contribute ideas that could help improve the success of the research thereby increasing the probability of success.

### Screening and enrolment of participants
Study recruitment opened on 17 April 2020. Study activities are as detailed below (also detailed in online supplemental file 2).

#### *Pre-screening*
We conducted a house-to-house enumeration (census) of all potential participants (children under 6 months of age) living in the target area and generated a study population database. New births are identified and reported by HSAs or registered at the health facilities' under-5 clinics throughout the trial recruitment period and added to the database. Research assistants visit all children approaching 6 months of age in the community. After taking written informed consent, the team reviews the child's medical history to determine potential eligibility. Eligible children are invited to visit a selected health facility for further screening.

#### *Screening*
On a predetermined day, the child, accompanied by their parent(s) or legal guardian, visit the selected health centre. HemoCue Hb301+ (Angelholm, Sweden) is used to assess capillary Hb level to exclude severely anaemic children. Additionally, anthropometric measurements: weight and length (used to calculate weight-for-length z-scores[42]) and MUAC are performed to exclude severely malnourished children. Severely anaemic and severely malnourished children are referred to the routine health system for further investigation and treatment.

#### *Baseline and enrolment (at 6 months of age (±14 days))*
A second, more detailed consent form for participation in the study is signed, and a study identification number is assigned. Randomisation is performed after the following baseline evaluations (see online supplemental file 2) are carried out and necessary information collected: demographic information, socioeconomic status (using the Malawi Demographic Health Survey standard of living scale), medical history, family care indicators[43]/index, feelings questionnaire,[44] cost survey (household out of pocket healthcare costs), food security questionnaire,[45] dietary diversity questionnaire,[46] developmental screening using Caregiver Reported Early Development Instrument (CREDI),[47] anthropometric measurements (weight, length, MUAC and head circumference) and anthropometric measurements for the mother (weight, height). A venous blood sample (3 mL), for Hb testing, determination of iron indices (serum – for ferritin, C reactive protein (CRP) and Arabinogalactan protein

(AGP)), thick and thin malaria slides for malaria microscopy testing (to be conducted at the completion of the trial) and filter paper for plasmodium PCR, is collected from the child. The mother is given educational information regarding the use of study medications, administration of the first dose of DP and instructions on how to administer day 2 and day 3 DP treatments, administration of the first dose of iron syrup, use of MNPs and advice on future visits.

### Subsequent study visits
#### Adherence assessment
Fortnightly 'home visits' by experienced research assistants to the participants' homes in the community are done to assess the following: (1) documentation of episodes of infection (rate, days affected), (2) recording any adverse events/effects, (3) documentation of details of unplanned hospital and healthcare facility attendance, for example, to a doctor, traditional healer, clinic or hospital, (4) provision of educational information to participants on the correct use of the study interventions and encouraging adherence to study interventions, (5) adherence monitoring (measured from parental questioning along with assessment of residual MNPs and syrup) and (6) advice on future visits.

#### Monthly DP issuance
The research assistants issue the investigational products monthly during a home visit. The following activities (including data collection) are undertaken: issuance of the study interventions according to study protocol and adherence monitoring (measured from parental questioning along with assessment of residual MNPs and syrup). In addition, unused investigational products are collected.

#### Study midline visit (at 12 months of age (-2 weeks or +6 weeks))
The participating child and their parent/legal guardian attend the health facility at the end of the 6 months intervention for developmental assessments using locally adapted Bayley Scales of Infant and Toddler Development, Wolke's Behavioural Rating (WBR) and Child Temperament using the Bates' Infant Characteristics Questionnaire. Anthropometric measurements are conducted on both the child and the mother. Venous blood (3 mL) for Hb testing, determination of iron indices, malaria microscopy (to be conducted at the completion of the trial) and malaria filter paper for future PCR analyses is collected from the child.

#### Monthly malaria infection testing
Research assistants visit the homes of participants once per month after completion of the midline visit for assessment of episodes of infection (rate, number of days affected) in the previous 1 month and documentation of unplanned hospital and/or healthcare facility attendance (details of any interim visit to doctor, clinical or hospital visits in the previous month) and record of healthcare usage.

#### Endline visit (at 18 months of age (−2 weeks or +6 weeks))
The procedures are similar to the midline visit described above and in online supplemental file 2.

#### Unscheduled visits
Mothers or legal guardians are asked to attend the research site/clinic when their child is symptomatically unwell. They are managed according to national standard treatment guidelines by a trained healthcare provider. Blood samples for malaria rapid diagnostic test (mRDT) and blood culture are taken if clinically indicated. The clinical diagnosis of the unscheduled visit is recorded in the participant's record and captured on the study's interim visit case report form (CRF) and adverse event form. If the child requires inpatient care, they are referred to the Chikwawa district hospital.

### Laboratory procedures
A capillary blood sample is collected by finger prick from all age-eligible participants to measure the Hb concentration by HemoCue (HemoCue 301+, Angelholm, Sweden) and *Plasmodium* infection by RDT (mRDTs); (SD Bioline Malaria Ag Pf/Pan, Standard Diagnostics) at baseline. In addition, up to 3 mL venous blood sample is collected from all enrolled participants at baseline, midline, endline and unscheduled visits (if necessary) for Hb testing, determination of iron indices (serum – for ferritin, CRP and AGP), thick and thin malaria slides for malaria microscopy testing and filter paper for *Plasmodium* PCR. Families are encouraged to permit the collection of venous blood for venous Hb and iron and inflammation biomarkers at all time points; however, if a family refuses venous blood collection or if collection was unsuccessful, data for the primary outcome and other non-laboratory secondary outcomes will still be collected.

### Data Safety and Monitoring Committee
An independent Data Safety and Monitoring Committee (DSMC) comprised of international experts in clinical trials, child health, infectious diseases, epidemiology and statistics was set-up to review regularly, safety data of the ongoing trial. The DSMC may recommend to the sponsor and investigators whether to continue, modify or terminate the trial on ethical grounds.

### Outcomes
#### The primary outcome
Cognitive development: assessed by the differences in Bayley-III Cognitive Composite Score (CogCS) between intervention arms measured after 6 months intervention (midline) and 12 months post-intervention (endline). The Bayley-III is a validated index of child development that can be administered in ~0.5–1 hour and comprises cognitive, motor and language development scales. Assessors are specially trained to administer the tests, and inter-rater reliability is established before starting data collection. In addition, we perform 10% interobserver testing to monitor inter-rater reliability of the measure over the study period.

## Secondary outcomes

Secondary outcomes include: motor and language development assessed by Bayley-III Motor (MotCS) and Language (LangCS) Composite Scores (+6 months, +12 months), behaviour assessed by WBR, temperament, growth assessed by weight-for-age z-score, length-for-age z-score, weight-for-length z-score, head circumference z-score, anaemia (Hb<11 g/dL), Hb concentration, iron deficiency (ferritin <12 ng/μL adjusted for inflammation assessed by CRP or AGP), ferritin concentration and iron deficiency anaemia.

Safety will be assessed by prevalence of malaria parasitaemia and CRP and inflammation (CRP>5 mg/L) at the end of the intervention and a further 6 months post-intervention; episodes of caregiver reported symptoms (eg, fever, diarrhoea, vomiting, cough) determined by the biweekly and monthly morbidity questionnaire; episodes of unplanned clinical visits and hospitalisations and deaths.

## Detection and reporting of adverse events and serious adverse events

Non-serious adverse events and serious adverse events (SAEs) are collected from the time consent is given until the participant completes the study (the final visit or withdrawal). These are identified through spontaneous reports by the participant, unplanned visits to the research site or any participating health centres, observation by the study staff and standard questioning at each visit (biweekly during the intervention period and monthly during the post-intervention period). All SAEs are reported to the sponsor and the appropriate ethics committee, whether or not they are considered causally related to the investigational products.

## Sample size

The primary objective will be met if either (1) MNPs+DP is superior to DP alone, or (2) iron+DP alone is superior to DP alone or (3) DP alone is superior to placebo on the Bayley-III CogCS immediately after completion of the intervention (midline). A previous study in 2853 Bangladeshi children showed that a 0.2 standardised mean difference (SMD) in Bayley-II Mental Development Index at 7 months translates to a 0.7 SMD at 18 months and a 1.2 SMD at 64 months age.[48] Therefore, preventing a loss of at least 0.2 SMD may produce magnified long-term developmental benefits as manifestations of brain dysfunction become more evident. A total sample size of 2168 infants (542 per group) will provide 90% power to detect an effect size of 0.25 SMD, assuming a 20% dropout and a two-sided alpha of 1.67% (Bonferroni adjustment for three primary between-group comparisons).

## Statistical analysis plan

A stand-alone statistical analysis plan will be detailed before breaking the blind. This plan will detail all principles and precise methods of analysis, including primary and secondary treatment effects of interest; alternative analyses if model assumptions do not hold; and additional analyses. Participants will be analysed according to their randomised treatment group for efficacy objectives and actual treatment group for safety objectives. The primary outcome of Bayley-III CogCS will be analysed using a mixed-model repeated measures analysis including in the model treatment group, time point and group by time point interaction. The stratification variables sex and site will be included in all models as main effects. In addition, the CREDI variable (collected at baseline) will be included as a main effect in the models for childhood development as it is considered prognostic for childhood development outcomes. Secondary repeated continuous outcomes (eg, growth outcomes) will be analysed similarly to Bayley-III CogCS. The outcome may be appropriately transformed before fitting the analysis model if considered skewed (eg, log-transformation for ferritin). Secondary repeated binary outcomes (eg, haematological outcomes) will be analysed using a log-binomial model and account for the repeated measures with a random intercept for participants. Safety (eg, AEs, infections, hospital attendance) will be summarised during the intervention and post-intervention periods separately and compared between the groups. Analysis of binary repeated outcome safety data (eg, inflammation) and continuous repeated outcome safety data (eg, CRP) will be analysed similarly to the secondary outcomes detailed above. Episode safety data (eg, caregiver reported diarrhoea episodes) will be analysed using negative binomial regression models. Subgroup analyses (eg, sex, site, iron deficiency) will be performed to explore the heterogeneity of the results independent of the study findings.

## Data management

Data collected from the subjects are recorded in digital CRFs using Research Electronic Data Capture (REDCap) software (Vanderbilt University, USA) on tablets. Relevant hard copy patient hospital files are scanned for reference and stored digitally, and securely. The trial is independently monitored by an external monitoring service. No interim analysis is planned.

## Ethics and dissemination

The National Health Science Research Committee, Malawi, and the Walter and Eliza Hall Institute of Medical Research Human Research Ethics Committee, Australia, approved this study. The Malawian Pharmacy and Medicines Regulatory Authority also approved the trial. Additionally, the protocol is registered with the Australian and New Zealand Clinical Trials Registry. Protocol modifications are reported to the ethics committees, regulatory authorities in Malawi, investigators, trial participants and trial registries. Written informed consent in the local language (see online supplemental file 3) is obtained from each participant's parent/legal guardian before conducting any study-related procedure. In the event where the participant's parent/legal guardian



is illiterate, the informed consent form is read out loudly to the potential participant's parent/legal guardian in the presence of an impartial witness. The potential participant's parent/legal guardian thumb-prints the consent form if they agree to participate in the study and the witness signs to indicate he/she witnessed the whole consenting process. The results will be presented and shared with the local community that hosted the research and to the international forum. We will publish in peer-reviewed scientific journals and report to relevant policymaking bodies like the Malawi Ministry of Health.

## Discussion

The IRMA trial will provide valuable evidence on the immediate and medium-term impacts of iron interventions (iron supplements and MNPs) and malaria prevention with DP on critical functional outcomes, most importantly, child cognitive development. These findings will complement high-quality data recently reported in South Asia to provide generalisable evidence for or against universal iron interventions across sub-Saharan Africa and other malaria-endemic settings.[10]

More than two-thirds of African children aged 6–59 months are anaemic. Preventing anaemia is considered important within the first 1000 days and aligns with WHO policy for children 6–23 months of age,[7] especially because the central nervous system is rapidly developing and is highly vulnerable. The prevalence of iron deficiency in children across Africa has been shown to increase from birth to approximately 1 year, with a peak prevalence at 9–12 months of age and then a decrease with increasing age. Iron interventions should thus target the critical under 1-year period.[49]

Recent clinical trials have indicated that the combination of intermittent preventive treatment of malaria in infants (IPTi) and malaria vaccine (RTS<S/AS01$_E$) provides much greater protection from malaria attacks in children compared to IPTi or RTS, S/AS01$_E$ given alone.[50] Combining iron interventions with IPTi, would thus be optimal for preventing both malaria and anaemia in young children in this setting.

The study's strengths include the design with cognitive development as the primary functional outcome, the large sample size with the power to detect a small effect size and the triple dummy design, which minimises the risk of bias. The presumed benefit of cognitive development is the key rationale for iron preventive interventions in children. Limitations include routine 6 monthly interval malaria microscopy testing, which is not a reliable test for the prevalence of malaria parasitaemia compared with a monthly interval. However, the strong surveillance systems on clinical symptoms (biweekly home visits and phone calls) ensured that all clinical malaria cases were picked up by ensuring that all children with fever and other symptoms could access hospital care and get a rapid malaria test done.

**Author affiliations**
[1]Training and Research Unit of Excellence (TRUE), Blantyre, Malawi
[2]The Micronutrient Forum, Healthy Mothers Healthy Babies Consortium, Washington DC, Washington, USA
[3]Department of Public Health, Kamuzu University of Health Sciences, Blantyre, Malawi
[4]Department of Health Promotion, Education, and Behaviour, University of South Carolina Arnold School of Public Health, Columbia, South Carolina, USA
[5]Department of Infectious Diseases at the Peter Doherty Institute of Infection and Immunity, The University of Melbourne, Melbourne, Victoria, Australia
[6]Centre for Epidemiology and Biostatistics, University of Melbourne School of Population and Global Health, Carlton, Victoria, Australia
[7]Population Health and Immunity Division, The Walter and Eliza Hall Institute of Medical Research, Parkville, Victoria, Australia
[8]International Centre for Diarrhoeal Disease Research, Dhaka, Bangladesh
[9]Diagnostic Haematology, The Royal Melbourne Hospital; and Clinical Haematology, Melbourne, Victoria, Australia
[10]Department of Medical Biology, Faculty of Medicine, Dentistry and Health Sciences, The University of Melbourne, Melbourne, Victoria, Australia

**Acknowledgements** We thank all the participants of the IRMA trial—including parents or legal guardians—especially during the challenging COVID-19 period. We thank the ethics and regulatory committees mentioned in the manuscript for their extensive review and approval of the study protocol and operational documents. We acknowledge the support from the Malawi Government, the Chikwawa District Health Office, including the health management teams of Chikwawa District Hospital, Makhwira, Mapelera, and Mfera Health Centres. We thank the Shire River East-and-West bank communities for hosting the study and continuously bearing with the extensive study procedures, especially the conduct of the two censuses to determine the extent of the source population. Finally, the study would not be possible without the support of over 100 staff members (in Malawi and Australia) who contribute their time and expertise to ensure its success. We say thank you to all.

**Contributors** MM drafted the initial study protocol. MM and GM supervised the trial's implementation and wrote the protocol paper's first draft. CCN and MDK coordinated the implementation of the study. JH, MV and LML planned for the measurement of the primary outcome. SB, RH and ARDM wrote the statistical analysis plan. MM and RA provided daily scientific input during the implementation of the trial. B-AB, KSP and S-RP conceived the trial, sought funding and provided senior scientific oversight in the conduct of the research. All authors read and approved the final manuscript.

**Funding** This work is supported by the National Health and Medical Research Council (NHMRC), Australia, grant numbers GNT1141185 and GNT1159151. S-RP is funded by NHMRC GNT1158696 and GNT2009047.

**Competing interests** None declared.

**Patient and public involvement** Patients and/or the public were involved in the design, or conduct, or reporting, or dissemination plans of this research. Refer to the Methods section for further details.

**Patient consent for publication** Not applicable.

**Provenance and peer review** Not commissioned; externally peer reviewed.

purpose, provided the original work is properly cited, a link to the licence is given, and indication of whether changes were made. See: https://creativecommons.org/licenses/by/4.0/.

**ORCID iDs**
Glory Mzembe http://orcid.org/0000-0002-7277-9987
Sabine Braat http://orcid.org/0000-0003-1997-3999
Rebecca Harding http://orcid.org/0000-0003-4267-716X
Alistair R. D. McLean http://orcid.org/0000-0003-3449-7142

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
