## [Reviewer comments · BMJ Open]

ARTICLE DETAILS

TITLE (PROVISIONAL)	Protocol for a randomised, multicentre, four-arm, double-blinded, placebo-controlled trial to assess the benefits and safety of Iron supplementation with Malaria chemoprevention to children in Malawi - IRMA trial
AUTHORS	Mwangi, Martin; Mzembe, Glory; Ngwira, Chikondi; Vokhiwa, Maclean; Kapulula, Mayamiko; Larson, Leila; Braat, Sabine; Harding, Rebecca; McLean, Alistair; Hamadani, Jena; Biggs, Beverley-Ann; Ataíde, Ricardo; Phiri, Kamija; Pasricha, Sant-Rayn

VERSION 1 – REVIEW

REVIEWER	Verhoef, Hans Wageningen UR, Division of Human Nutrition and Health I have previously collaborated and co-authored papers with some of the authors of the present paper, and I am also currently collaborating with them in a separate research project.
REVIEW RETURNED	28-Nov-2022

GENERAL COMMENTS	This trial is generally well-designed and addresses a new approach to solve an widespread and urgent health problem (anaemia in infants). The paper is generally clear and reads well. My main issue concerns the analysis of adverse events (point 7 below). 1. L. 65-68 'Trial setting': The authors may be overselling the generalisability of the trial results. Presumably, infants in the study are exposed to highly endemic malaria conditions. Results may be different in areas with lower endemicity (and, hence, with lower immunity derived from repeated or chronic exposure to infection) or areas where Plasmodium vivax is a major or the predominant malaria species (e.g., large parts of Ethiopia)2. L. 85-86 'The trial is powered to detect a small but clinically relevant effect': this statement is too broad and does not necessarily apply to all outcomes. I agree that the sample size is large for a trial of this nature, but the ability to detect small effects depends on the variance of the outcome.3. The trial objectives should be listed separately, not as part of the 'Methods and analysis' section.4. L. 204-206, primary objective: The wording seems somewhat ambiguous. Please clarify if the first part ('if the efficacy of iron supplements or MNPs (containing iron) given with malaria chemoprevention is superior to malaria chemoprevention alone') also will be measured with the Bayley-III Cognitive Composite Scores at the end of the 6-month intervention period as the primary outcome.5. L. 199-200 'and determine whether adding IPT diminishes potential risks [of iron supplementation]': This is not listed as a trial objective, and I believe that it is also not possible to investigate with
--

	the design proposed: it would require an additional arm in which children receive iron but not IPT. 6. As foreseen, the trial interventions covers the age range 6months to 1 year. The authors should clarify the choice for this age range and duration of the interventions. In the population studied, malaria probably peaks within this age range, but most malaria cases occur at a later age (1-3 years). Likewise, iron deficiency probably peaks at a slightly older age. I presume that the author have selected this age range with a view that future interventions can possibly be integrated within existing health care interventions such as vaccination programmes. If so, this should be clarified. 7. Analysis of adverse events: I find it difficult to reconcile the safety objectives, the surveillance system and the statistical analysis proposed. Safety of iron will be investigated by comparing groups regarding 'the risk of malaria and other morbidities' (L. 223) and 'clinical malaria, parasitaemia and malaria-specific clinic visits and all-cause sick visits during and after interventions (L. 226-227). For adverse events of interest (fever, respiratory disease, diarrhoea, clinic/hospital attendance, Table 1; constipation, vomiting and death, L. 427), the team will record the rate and number of days affected during scheduled, fortnightly home visits (in addition to monitoring of SEs/SAEs). In the analysis, safety will be assessed using log-binomial regression (L. 461). The Abstract states that malaria incidence will be measured (which is impossible by log-binomial regression). a. The log-binomial model requires the observations to be independent of each other, i.e., the observations should not come from repeated measurements. This is clearly not the case in the current study: the disease status on any given day is likely to be dependent on the disease status of previous or subsequent days. b. One limitation of the log-binomial model is that it often fails to converge because the instantaneous slope of the likelihood may not be zero on the boundary of the parameter space (https://www.ncbi.nlm.nih.gov/pmc/articles/PMC2292207/). What approaches will the authors take if this occurs? c. Considering the assumption of independence in log-binomial models, the authors may wish to consider beta regression (with zero-inflation as appropriate) as an alternative method to estimate the proportion of days affected. d. For malaria rates, analysis should be based on a definition of a new episode and should take into account the possibility of recurrent episodes. For example, an episode could be defined as one or more successive days with fever defined as axillary temperature >37.5 degrees Celsius plus a positive rapid dipstick test, separated by at least 7 successive days without fever (in that case, 7 days post-episode should be discounted from the total observation time because children are not at risk of a new episode because recurrence of fever would be interpreted as part of the previous episode). Depending on the assumptions, analysis can be done using count models (with zero-inflation and using an offset as appropriate) or survival analysis with recurrent events. To improve specificity, it would be desirable that such case definitions incorporate a quantitative measure of parasite load because, in highly endemic areas, many children within the age range studied are likely to be infected without symptoms. Similar issues are important for other adverse event outcomes (e.g., for clinic attendance, would a follow-up visit to monitor response to treatment be considered a separate event?).
--	---

	e. With regards to malaria microscopy, it is not clear how much time will occur between blood collection, staining and slide reading, which heavily influences the reliability of results. If possible, the authors should use quantitatively estimate DNA content from qPCR tests, or perhaps even better, results from parasite-specific DNA content by PCR in serum (https://pubmed.ncbi.nlm.nih.gov/25344520/) or serum concentration of HRP-2 concentration by quantitative ELISA. f. With regards to diarrhoea, it will be difficult to interpret results if mothers have a different concept of diarrhoea than the research team. Did you use pictorial charts (e.g., Bristol stool form scale) during the field work? 8. The Discussion section is very short and more a summary than a discussion. I encourage the authors to put their trial in a broader perspective, and to also discuss trial limitations. For example, with regards to dihydroartemisinin-piperaquine, what is the current state of Plasmodium resistance to piperaquine in Africa, and how in the authors' view will DP resistance affect possible roll-out and uptake of DP intervention in the future; how is the current update of the RTS,S malaria vaccine in Malawi, to what extent is there an overlap in coverage between the vaccine and DP chemoprevention, and how does this vaccine affect the efficacy and safety of the interventions investigated; do the authors have a vision about interventions in children > 1 year. Assumptions and limitations with respect to safety evaluations also deserve more attention (see also point 7 above). 9. P. 14: please state the target dose and the actual method of administration of DP. I presume that 20/160mg refers to contents of paediatric tablets. The recommended dose is 2/16 mg/kg, once daily for 3 days. How will this be dosed by weight (please provide a table, if necessary as an Annex)? Are dosages rounded to the nearest quarter tablet? 10. L. 338-342: Please provide references for various scales and instruments mentioned (socioeconomic status; family care indicators/index, FCI; feelings questionnaire; food security; dietary diversity questionnaire; developmental screening using Caregiver Reported Early Development Instrument, CREDI). 11. L. 447: The authors propose a Bonferroni adjustment for three primary between-group comparisons. Although I recognise that this is a controversial area, I would disagree for reasons given by Schultz and Grimes (https://pubmed.ncbi.nlm.nih.gov/15866314/). Also, would the authors consider that Bonferroni correction would be needed if the three between-group comparisons would be done in separate trials? (I hope not.)
--	--

REVIEWER	Andersen, Chris World Bank
REVIEW RETURNED	05-Dec-2022

GENERAL COMMENTS	The research question is important and the methods are appropriate. The authors have well justified the rationale for the study and provided a clear description of the study procedures. The results of the study will be important to inform the benefits and risks of iron supplementation on functional outcomes among children in malaria-endemic regions. I have only two comments for consideration by the editors and authors:
--

	1. My understanding is that the editors of BMJ Open intend for protocols to be published for "planned or ongoing" studies. A date is specified for when recruitment of the study began (17 April 2020, line 317). Given that it has been about two and half years since that date, I believe it would be important to specify the projected date at which data collection will be complete (so as to confirm that the study is still eligible for publication as a protocol). 2. I am uncomfortable with the fact that one study arm is not provided any malaria chemoprevention. While I acknowledge that the study has been approved by two ethical review boards, I do wonder how withholding malaria chemoprevention from study participants can be justified. The authors state that the placebo group allows for the "evaluation of the effect of malaria chemoprevention alone on anaemia and child development." However, the authors also cite prior evidence to demonstrate that malaria chemoprevention reduces malaria incidence (lines 167-169) and improves cognitive performance (lines 162-163). Given that there are clear benefits associated with chemoprevention, I think the authors should provide further explanation of their view on how the study meets a standard of " equipoise " for randomization to placebo (i.e. no chemoprevention and no iron).
--	---

REVIEWER	Tchum, Kofi Kintampo Health Research Centre, Clinical Laboratory
REVIEW RETURNED	15-Dec-2022

GENERAL COMMENTS	This manuscript was well written and the science was good. However, I have few comments and suggestions:  1. Please clearly state the meaning of "IRMA". Page 3 line 26. 2. Page 7, Lines 129 -131 should be "The seminal Pemba RCT in Tanzania was stopped early due to increased death, hospitalisation and malaria in the children with iron replete receiving iron compared with placebo and called for revisions of universal iron supplementation recommendations rather than "The seminal Pemba RCT in Tanzania was stopped early due to increased death, hospitalisation and malaria in the children receiving iron compared with placebo and called for revisions of universal iron supplementation recommendations". 3. Page 13, Lines 247 -248 should be Children are randomly assigned to a study group only if they meet all the inclusion and none of the exclusion criteria and not "Children are randomly assigned to a study intervention group only if they meet all the inclusion and none of the exclusion criteria". 4. Generally, ferritin levels are assayed with levels of transferrin receptor. Please this manuscript only the ferritin was determined. Any reason? 5. Hemocue (line 328) and RedCap (Line 471) should have name of manufacturer and country or place of origin. 6. Page 26, Lines 471 -482. In case if the caregiver was an illiterate. How that individual will be consented was not captured. 7. The rationale for taken anthropometric measurement of caregivers was not stated in the manuscript.
--

VERSION 1 – AUTHOR RESPONSE

Reviewer: 1

Dr. Hans Verhoef, Wageningen UR

Comments to the Author:

Manuscript ID bmjopen-2022-069011

Protocol for a randomised, multicentre, four-arm, double-blinded, placebo-controlled trial to assess the benefits and safety of Iron supplementation with Malaria chemoprevention to children in Malawi - IRMA trial. (Mwangi et al.).

This trial is generally well-designed and addresses a new approach to solve an widespread and urgent health problem (anaemia in infants). The paper is generally clear and reads well. My main issue concerns the analysis of adverse events (point 7 below).

1. L. 65-68 'Trial setting': The authors may be overselling the generalisability of the trial results. Presumably, infants in the study are exposed to highly endemic malaria conditions. Results may be different in areas with lower endemicity (and, hence, with lower immunity derived from repeated or chronic exposure to infection) or areas where *Plasmodium vivax* is a major or the predominant malaria species (e.g., large parts of Ethiopia).'

We agree with the reviewer and have limited the generalizability of the trial results to highly malaria-endemic parts of Sub-Saharan Africa (Line 65).

2. L. 85-86 'The trial is powered to detect a small but clinically relevant effect': this statement is too broad and does not necessarily apply to all outcomes. I agree that the sample size is large for a trial of this nature, but the ability to detect small effects depends on the variance of the outcome. We have specified that the sample size was based on the primary outcome which is child developmental outcomes.

(Lines 85-88): "The trial is powered to detect a small but clinically relevant effect for the primary outcome of child cognitive development; the use of a post-intervention follow-up period enables the assessment of both the immediate and medium-term impact of the intervention."

3. The trial objectives should be listed separately, not as part of the 'Methods and analysis' section. We agree with the reviewer and have listed the objectives separately and not as part of the 'Methods and analysis' section (Line 203).

4. L. 204-206, primary objective: The wording seems somewhat ambiguous. Please clarify if the first part ('if the efficacy of iron supplements or MNPs (containing iron) given with malaria chemoprevention is superior to malaria chemoprevention alone') also will be measured with the Bayley-III Cognitive Composite Scores at the end of the 6-month intervention period as the primary outcome.

Yes, both the efficacy of 1) iron supplements or MNPs (containing iron) given with malaria chemoprevention compared with malaria chemoprevention alone; and 2) malaria chemoprevention alone versus placebo will be measured with the Bayley-III cognitive Composite Scores at the end of the 6-month intervention period as the primary outcome.

5. L. 199-200 'and determine whether adding IPT diminishes potential risks [of iron supplementation]': This is not listed as a trial objective, and I believe that it is also not possible to investigate with the design proposed: it would require an additional arm in which children receive iron but not IPT.

We agree with the reviewer that there is no group that is receiving iron only. However, there are two groups; receiving IPT only and another group receiving placebo. We feel that if the risk of infectious disease is higher in the iron + IPT group than the placebo group, and higher than the IPT alone group, the excess risk could relatively be attributed to iron supplementation. It would be unethical to include an iron-only arm as there is sufficient evidence that iron supplementation alone increases infectious disease risk in this setting. Thus, evidence of no increased infectious disease risk in the iron + IPT arm could be relatively attributed to the addition of malaria IPT.

6. As foreseen, the trial interventions covers the age range 6 months to 1 year. The authors should clarify the choice for this age range and duration of the interventions. In the population studied, malaria probably peaks within this age range, but most malaria cases occur at a later age (1-3 years). Likewise, iron deficiency probably peaks at a slightly older age. I presume that the author have selected this age range with a view that future interventions can possibly be integrated within existing health care interventions such as vaccination programmes. If so, this should be clarified.

The trial was designed with a focus on improving WHO policy for children 6-23m of age; which is considered the peak period for iron deficiency anaemia and when the central nervous system is rapidly developing and highly vulnerable: thus, a time when preventing anaemia is considered important within the first 1000 days. Despite most malaria cases peaking at a later age (1-3 years), the prevalence of iron deficiency in children across Africa has been shown to increase from birth to approximately 1 year, with a peak prevalence at 9-12 months of age, and then decrease with increasing age, thus iron interventions should target the critical under 1 year period (<https://bmcmmedicine.biomedcentral.com/articles/10.1186/s12916-020-1502-7>).

The period 6 months to 1 year is when complementary foods replace breastfeeding; thus, the earliest time this could be tried (MNPs are added to semi-solid meals). Additionally, the interventions can be linked to vaccination schedules.

7. Analysis of adverse events: I find it difficult to reconcile the safety objectives, the surveillance system and the statistical analysis proposed. Safety of iron will be investigated by comparing groups regarding 'the risk of malaria and other morbidities' (L. 223) and 'clinical malaria, parasitaemia and malaria-specific clinic visits and all-cause sick visits during and after interventions (L. 226-227). For adverse events of interest (fever, respiratory disease, diarrhoea, clinic/hospital attendance, Table 1; constipation, vomiting and death, L. 427), the team will record the rate and number of days affected during scheduled, fortnightly home visits (in addition to monitoring of SEs/SAEs). In the analysis, safety will be assessed using log-binomial regression (L. 461). The Abstract states that malaria incidence will be measured (which is impossible by log-binomial regression).

Thanks to the reviewer for drawing our attention to this. We are collecting a broad swathe of safety data which includes binary outcomes (e.g. inflammation at midline and endline); continuous outcomes (e.g. CRP at midline and endline) and incidence outcomes (e.g. diarrhoea episodes). The analysis section incorrectly omitted the models that will be used for the continuous outcomes and incidence outcomes, this has now been amended (Lines 452-475). Further details have been provided in response to specific points below.

a. The log-binomial model requires the observations to be independent of each other, i.e., the observations should not come from repeated measurements. This is clearly not the case in the current study: the disease status on any given day is likely to be dependent on the disease status of previous or subsequent days.

The log-binomial model will be used for the binary safety outcomes such as inflammation at midline and endline. A single level log-binomial model does assume that all observations are independent (including observations from the same individual). However, we are fitting a multilevel model with a random intercept for participants, to account for repeated measurements.

b. One limitation of the log-binomial model is that it often fails to converge because the instantaneous slope of the likelihood may not be zero on the boundary of the parameter space (<https://www.ncbi.nlm.nih.gov/pmc/articles/PMC2292207/>). What approaches will the authors take if this occurs?

The statistical analysis plan will outline alternative models that will be fit in the event of non-convergence.

c. Considering the assumption of independence in log-binomial models, the authors may wish to consider beta regression (with zero-inflation as appropriate) as an alternative method to estimate the proportion of days affected.

For binary outcomes with repeated measurements we are fitting a multilevel model with a random intercept for participants, to account for repeated measurements. We will not be using log-binomial models for continuous or incidence outcomes where we will be using MMRM and negative binomial regression models respectively.

d. For malaria rates, analysis should be based on a definition of a new episode and should take into account the possibility of recurrent episodes. For example, an episode could be defined as one or more successive days with fever defined as axillary temperature >37.5 degrees Celsius plus a positive rapid dipstick test, separated by at least 7 successive days without fever (in that case, 7 days post-episode should be discounted from the total observation time because children are not at risk of a new episode because recurrence of fever would be interpreted as part of the previous episode). Depending on the assumptions, analysis can be done using count models (with zero-inflation and using an offset as appropriate) or survival analysis with recurrent events. To improve specificity, it would be desirable that such case definitions incorporate a quantitative measure of parasite load because, in highly endemic areas, many children within the age range studied are likely to be infected without symptoms. Similar issues are important for other adverse event outcomes (e.g., for clinic attendance, would a follow-up visit to monitor response to treatment be considered a separate event?).

To avoid counting the same episode of clinical malaria twice, any episodes recorded within 28 days of a previously recorded event of clinical malaria will not be counted as a new event. Similarly, a period of 2 weeks without symptoms will be used to avoid counting diarrhoeal and respiratory disease episodes twice. For clinic attendance, follow-up visit to monitor response to treatment, follow-up visits due to non-resolving or worsening symptoms or readmissions will not be considered as separate events. Analysis of incidence data will be conducted using negative binomial regression. Where a time period exists where it is impossible by definition for another episode to occur, this time period will be removed from the exposure time of that individual. Further details of the definitions of each safety outcome, and the calculation of exposure time will be detailed in the Statistical Analysis Plan that will be finalized prior to database lock.

e. With regards to malaria microscopy, it is not clear how much time will occur between blood collection, staining and slide reading, which heavily influences the reliability of results. If possible, the authors should use quantitatively estimate DNA content from qPCR tests, or perhaps even better, results from parasite-specific DNA content by PCR in serum (<https://pubmed.ncbi.nlm.nih.gov/25344520/>) or serum concentration of HRP-2 concentration by quantitative ELISA.

Filter paper for PCR was collected in all infants at routinely scheduled visits at baseline, at the end of the 6 months intervention and a further 6 months post intervention, but will not be included as part of the primary analysis of the trial. Slide staining was done immediately after sample collection but we delayed slide reading until after the infants exited the study to avoid being forced to withhold clinical information.

f. With regards to diarrhoea, it will be difficult to interpret results if mothers have a different concept of diarrhoea than the research team. Did you use pictorial charts (e.g., Bristol stool form scale) during the field work?

No pictorial charts were used. We applied the standard WHO definition of a diarrhoea episode: the passage of three or more loose or liquid stools per day (<https://www.who.int/news-room/factsheets/detail/diarrhoeal-disease>).

8. The Discussion section is very short and more a summary than a discussion. I encourage the authors to put their trial in a broader perspective, and to also discuss trial limitations. For example, with regards to dihydroartemisinin-piperazine, what is the current state of Plasmodium resistance to piperazine in Africa, and how in the authors' view will DP resistance affect possible roll-out and uptake of DP intervention in the future; how is the current update of the RTS,S malaria vaccine in Malawi, to what extent is there an overlap in coverage between the vaccine and DP chemoprevention, and how does this vaccine affect the efficacy and safety of the interventions investigated; do the authors have a vision about interventions in children > 1 year. Assumptions and limitations with respect to safety evaluations also deserve more attention (see also point 7 above).

This has been revised accordingly.

Lines 504-524:

More than two-thirds of African children aged 6-59 months are anaemic. Preventing anaemia is considered important within the first 1000 days and aligns with WHO policy for children 6-23m of age,⁴⁹ especially because the central nervous system is rapidly developing and is highly vulnerable. The prevalence of iron deficiency in children across Africa has been shown to increase from birth to approximately one year, with a peak prevalence at 9-12 months of age, and then a decrease with increasing age. Iron interventions should thus target the critical under one year period.⁵⁰

Recent clinical trials have indicated that the combination of Intermittent preventive treatment of malaria in infants (IPTi) and malaria vaccine (RTS

The study's strengths include the design with cognitive development as the primary functional outcome, the large sample size with the power to detect a small effect size, and the triple dummy design, which minimizes the risk of bias. The presumed benefit of cognitive development is the key rationale for iron preventive interventions in children. Limitations include routine six monthly interval malaria microscopy testing, which is not a reliable test for the prevalence of malaria parasitaemia compared to a monthly interval. However, the strong surveillance systems on clinical symptoms (biweekly home visits and phone calls) ensured that all clinical malaria cases were picked by ensuring that all children with fever and other symptoms could access hospital care and get a rapid malaria test done.

9. P. 14: please state the target dose and the actual method of administration of DP. I presume that 20/160mg refers to contents of paediatric tablets. The recommended dose is 2/16 mg/kg, once daily for 3 days. How will this be dosed by weight (please provide a table, if necessary as an Annex)? Are dosages rounded to the nearest quarter tablet?

The recommended dose of 2/16mg/Kg once daily for 3 days was used. The dose was calculated based on the infant's weight on enrolment. The dosage was rounded to the nearest half a tablet. A child maintained the same dosage throughout the 6 months intervention period (Line 279).

10. L. 338-342: Please provide references for various scales and instruments mentioned (socioeconomic status; family care indicators/index, FCI; feelings questionnaire; food security; dietary diversity questionnaire; developmental screening using Caregiver Reported Early Development Instrument, CREDI).

These have been added accordingly (Lines 339-343).

Additionally, we have noticed that an important detail of the statistical analysis related to CREDI had been accidentally omitted, and have included it in line 462-464:

"In addition, the CREDI variable (collected at baseline) will be included as a main effect in the models for childhood development as it is considered prognostic for childhood development outcomes."

11. L. 447: The authors propose a Bonferroni adjustment for three primary between-group comparisons. Although I recognise that this is a controversial area, I would disagree for reasons given by Schultz and Grimes (<https://pubmed.ncbi.nlm.nih.gov/15866314/>). Also, would the authors

consider that Bonferroni correction would be needed if the three between-group comparisons would be done in separate trials? (I hope not.)

We agree that this is a controversial area, and are sympathetic to the reviewer's perspective.

However, the trial was reviewed and given approval by ethics committees based on a sample size calculation that utilised a Bonferonni adjustment and as such we do not think it is appropriate to alter this aspect of the study design. We will report the divergence (unadjusted) p values alongside the decision (adjusted) p values to allow readers who disagree with the Bonferroni approach to assess the evidence using unadjusted p values if they wish.

Reviewer: 2

Dr. Chris Andersen, World Bank

Comments to the Author:

The research question is important and the methods are appropriate. The authors have well justified the rationale for the study and provided a clear description of the study procedures. The results of the study will be important to inform the benefits and risks of iron supplementation on functional outcomes among children in malaria-endemic regions.

I have only two comments for consideration by the editors and authors:

1. My understanding is that the editors of BMJ Open intend for protocols to be published for "planned or ongoing" studies. A date is specified for when recruitment of the study began (17 April 2020, line 317). Given that it has been about two and half years since that date, I believe it would be important to specify the projected date at which data collection will be complete (so as to confirm that the study is still eligible for publication as a protocol).

Data collection in the field is complete, but the data analysis (including lab analysis/ sample interpretation) remains incomplete. The trial database is not locked as there is still blinded lab analysis, data validation and cleaning taking place. We plan to have the database locked in at the earliest April 2023.

2. I am uncomfortable with the fact that one study arm is not provided any malaria chemoprevention. While I acknowledge that the study has been approved by two ethical review boards, I do wonder how withholding malaria chemoprevention from study participants can be justified. The authors state that the placebo group allows for the "evaluation of the effect of malaria chemoprevention alone on anaemia and child development." However, the authors also cite prior evidence to demonstrate that malaria chemoprevention reduces malaria incidence (lines 167-169) and improves cognitive performance (lines 162-163). Given that there are clear benefits associated with chemoprevention, I think the authors should provide further explanation of their view on how the study meets a standard of "equipoise" for randomization to placebo (i.e. no chemoprevention and no iron).

Malaria chemoprevention is not given routinely in this setting. Thus, the placebo arm is the current standard of care in the country and no treatment is being withheld from the participants.

Reviewer: 3

Dr. Kofi Tchum, Kintampo Health Research Centre

Comments to the Author:

Reviewer's Comments and suggestions

This manuscript was well written and the science was good. However, I have few comments and suggestions:

1. Please clearly state the meaning of "IRMA". Page 3 line 26.

Benefits and safety of Iron supplementation with malaria chemoprevention to children in Malawi (IRMA) - A randomised controlled trial.

2. Page 7, Lines 129 -131 should be “The seminal Pemba RCT in Tanzania was stopped early due to increased death, hospitalisation and malaria in the children with iron replete receiving iron compared with placebo and called for revisions of universal iron supplementation recommendations rather than “The seminal Pemba RCT in Tanzania was stopped early due to increased death, hospitalisation and malaria in the children receiving iron compared with placebo and called for revisions of universal iron supplementation recommendations”.

This has been revised accordingly (Line 130).

3. Page 13, Lines 247 -248 should be Children are randomly assigned to a study group only if they meet all the inclusion and none of the exclusion criteria and not “Children are randomly assigned to a study intervention group only if they meet all the inclusion and none of the exclusion criteria”.

This has been revised accordingly (Line 249).

4. Generally, ferritin levels are assayed with levels of transferrin receptor. Please this manuscript only the ferritin was determined. Any reason?

In this study, we intend to use Ferritin + CRP (Ferritin adjusted for inflammation).

5. Hemocue (line 328) and RedCap (Line 471) should have name of manufacturer and country or place of origin.

This has been added accordingly (Line 329 and line 478).

6. Page 26, Lines 471 - 482. In case if the caregiver was an illiterate. How that individual will be consented was not captured.

This has been added accordingly.

Lines 489-493: “In the event where the participant is illiterate, the informed consent form would be read out loudly to the potential participant in the presence of an impartial witness. The potential participant would thumbprint if they agree to participate in the study and the witness would sign to indicate the witnessed the whole consenting process”.

7. The rationale for taken anthropometric measurement of caregivers was not stated in the manuscript.

One of the study’s objectives is to assess the effect of the intervention on an infant’s physical growth. An infant’s physical growth is correlated with maternal physical stature hence taking the anthropometric measurement of the caregivers. From experience from our other clinical trials conducted in this setting,, >95% of the infants come to the clinic with their mothers. On our data collection tool, it is specified on the relationship of the caretaker to the child to capture this information accurately. The data also provides opportunities for use in future secondary studies.

VERSION 2 – REVIEW

REVIEWER	Verhoef, Hans Wageningen UR, Division of Human Nutrition and Health
REVIEW RETURNED	04-May-2023
GENERAL COMMENTS	Regarding my previous point 5: I accept the argument that an iron-only arm is ethically unacceptable in this population, and I agree that an excess risk of infectious disease as determined by comparison of ‘iron + IPT’ versus ‘IPT only’ would be attributable

	to iron. I persist in my belief, however, that the trial design does not allow an investigation to determine whether adding IPT diminishes potential risks of iron supplementation. If it turns out that the 'iron + IPT' and 'placebo' groups have similar risk of infectious disease, one cannot attribute such a result to IPT having suppressed excess risk of infectious disease due to iron. Regarding my previous point 7a: Accepted insofar (repeated) survey data are concerned
--	--

REVIEWER	Andersen, Chris World Bank
REVIEW RETURNED	28-Feb-2023

GENERAL COMMENTS	Thanks to the authors for their replies. I remain personally uncomfortable with the withholding of a proven intervention (i.e. malaria chemoprevention) from one arm of study participants - while it is being given to the other three arms - on the basis that this treatment is not typically available to this economically deprived population. However, I understand that this study has received ethical approval, so I will not obstruct the publication of this otherwise well-designed and useful study protocol.
---

REVIEWER	Tchum, Kofi Kintampo Health Research Centre, Clinical Laboratory
REVIEW RETURNED	21-Feb-2023

GENERAL COMMENTS	This manuscript is very remarkable and the science is impressive.
---

VERSION 2 – AUTHOR RESPONSE

Reviewer: 3

Dr. Kofi Tchum, Kintampo Health Research Centre

Comments to the Author:

This manuscript is very remarkable and the science is impressive

We thank the reviewer for this comment.

Reviewer: 2

Dr. Chris Andersen, World Bank

Comments to the Author:

Thanks to the authors for their replies. I remain personally uncomfortable with the withholding of a proven intervention (i.e. malaria chemoprevention) from one arm of study participants - while it is being given to the other three arms - on the basis that this treatment is not typically available to this economically deprived population. However, I understand that this study has received ethical approval, so I will not obstruct the publication of this otherwise well-designed and useful study protocol.

We thank the reviewer for this comment

Reviewer: 1

Dr. Hans Verhoef, Wageningen UR

Comments to the Author:

Regarding my previous point 5: I accept the argument that an iron-only arm is ethically unacceptable in this population, and I agree that an excess risk of infectious disease as determined by comparison of 'iron + IPT' versus 'IPT only' would be attributable to iron. I persist in my belief, however, that the trial design does not allow an investigation to determine whether adding IPT diminishes potential risks of iron supplementation. If it turns out that the 'iron + IPT' and 'placebo' groups have similar risk of infectious disease, one cannot attribute such a result to IPT having suppressed excess risk of infectious disease due to iron.

Regarding my previous point 7a: Accepted insofar (repeated) survey data are concerned.

We agree with the reviewer that it will not be possible to investigate if adding IPT diminishes the potential risks of iron supplementation without an additional arm in which children iron but not IPT.

Lines 189-190: The IRMA trial will assess functional health outcomes from iron supplementation and identify the ideal mode of iron delivery (MNPs versus iron syrup/drops). It will also allow evaluation of the effect of malaria chemoprevention alone on anaemia and child development.